# Molecular Biomarkers in Glioblastoma: A Systematic Review and Meta-Analysis

**DOI:** 10.3390/ijms23168835

**Published:** 2022-08-09

**Authors:** Heena Sareen, Yafeng Ma, Therese M. Becker, Tara L. Roberts, Paul de Souza, Branka Powter

**Affiliations:** 1Centre for Circulating Tumour Cell Diagnostics and Research, Ingham Institute for Applied Medical Research, Liverpool, NSW 2170, Australia; 2South-Western Clinical School, University of New South Wales, Liverpool, NSW 2170, Australia; 3School of Medicine, Western Sydney University, Campbelltown, NSW 2560, Australia; 4Liverpool Hospital, Liverpool, NSW 2170, Australia

**Keywords:** glioblastoma, prognostic biomarkers, systematic review, meta-analysis

## Abstract

Background: Glioblastoma (GBM) is a highly aggressive cancer with poor prognosis that needs better treatment modalities. Moreover, there is a lack of reliable biomarkers to predict the response and outcome of current or newly designed therapies. While several molecular markers have been proposed as potential biomarkers for GBM, their uptake into clinical settings is slow and impeded by marker heterogeneity. Detailed assessment of prognostic and predictive value for biomarkers in well-defined clinical trial settings, if available, is scattered throughout the literature. Here we conducted a systematic review and meta-analysis to evaluate the prognostic and predictive significance of clinically relevant molecular biomarkers in GBM patients. Material and methods: A comprehensive literature search was conducted to retrieve publications from 3 databases (Pubmed, Cochrane and Embase) from January 2010 to December 2021, using specific terms. The combined hazard ratios (HR) and confidence intervals (95% CI) were used to evaluate the association of biomarkers with overall survival (OS) in GBM patients. Results: Twenty-six out of 1831 screened articles were included in this review. Nineteen articles were included in the meta-analyses, and 7 articles were quantitatively summarised. Fourteen studies with 1231 GBM patients showed a significant association of *MGMT* methylation with better OS with the pooled HR of 1.66 (95% CI 1.32–2.09, *p* < 0.0001, random effect). Five studies including 541 GBM patients analysed for the prognostic significance of *IDH1* mutation showed significantly better OS in patients with *IDH1* mutation with a pooled HR of 2.37 (95% CI 1.81–3.12; *p* < 0.00001]. Meta-analysis performed on 5 studies including 575 GBM patients presenting with either amplification or high expression of EGFR gene did not reveal any prognostic significance with a pooled HR of 1.31 (95% CI 0.96–1.79; *p* = 0.08). Conclusions: *MGMT* promoter methylation and *IDH1* mutation are significantly associated with better OS in GBM patients. No significant associations were found between *EGFR* amplification or overexpression with OS.

## 1. Introduction

Glioblastoma (GBM), the most common and aggressive form of brain cancer, has an overall 5-year survival of only 7% [1]. Based on clinical presentation, GBM is classified into two different categories: Primary GBM accounts for approximately 90% of GBM cases, arises de-novo, and is more common in elderly patients [2]. It is characterised by distinct molecular alterations that include gene amplification of epidermal growth factor receptor (*EGFR*), overexpression of EGFR protein and loss of the tumour suppressor gene phosphatase and tensin homologue (*PTEN*) [2]. Secondary GBM accounts for approximately 10% of cases, is associated with younger patient age, and arises from lower grade precursors. Secondary GBM has better prognosis and typically carries mutations in isocitrate dehydrogenase 1 (*IDH1*) and Tumour protein 53 (*TP53*) genes [2].

Standard GBM treatment involves surgical resection, where the extent of resectable tumour is dictated by risk to the patient, as the tumour often infiltrates essential parts of the brain. Following surgical treatment, patients are treated with radiation and concomitant temozolomide (TMZ), then adjuvant TMZ to target remaining tumour cells [3]. Recurrence is, on average, observed 7–10 months post treatment [4]. The median overall survival (OS) of GBM patients is 12–14 months, even after treatment with TMZ and radiation, primarily due to the invasive nature of the cancer and resistance to therapies [4].

To date, few molecular biomarkers have been discovered. These include O^6^-methylguanine DNA methyltransferase (*MGMT*) promoter methylation [5], Isocitrate dehydrogenase 1 (*IDH1*) mutation [6], mutations in the promoter region of the telomerase reverse transcriptase (*TERT*) gene [7], and amplification and/or overexpression of EGFR [8]. These markers have shown potential to predict the survival outcomes and treatment response in GBM patients. *MGMT* promoter methylation is known as a positive prognostic biomarker for patients treated with alkylating agents such as TMZ [5]. The *MGMT* gene encodes a DNA repair protein, which reverses DNA alkylation [9]. *MGMT* promoter methylation reduces its expression, thereby rendering cells more vulnerable to alkylating agents [9]. *IDH1* mutation (R132H) is also considered a favourable prognostic biomarker for GBM patients. It is more common in younger patients (18–45 years) and more frequent in secondary GBM (~73%), while rare in primary GBM (~3.7%) [10,11]. *EGFR* amplification and overexpression have been implicated as prognostic and predictive biomarkers [8]. Various EGFR targeting tyrosine kinase inhibitors (TKIs), such as gefitinib, erlotinib, and afatinib, have been trialled as targeted therapies for GBM [12].

These potential molecular biomarkers have value for GBM patient management or are informative in the context of standard of care TMZ and radiation treatment. However, for a subset of patients, the outcome is not well predicted by these markers, and may be comparably better or worse than predicted. This highlights the need to investigate other biomarkers associated with prognosis and response to treatment, particularly for newer treatment modalities. For instance, VEGF is proposed to drive angiogenesis and tumourigenesis due to its aberrant expression in GBM patients [13], and is therefore an attractive therapeutic target. Bevacizumab, a humanised antibody to vascular endothelial growth factor (VEGF), approved for recurrent GBM in many countries, was also trialled in newly diagnosed patients. However, no survival benefit was reported in the newly diagnosed GBM patients receiving bevacizumab in addition to standard of care [14]. Many studies have proposed various pharmacodynamic, prognostic and predictive biomarkers to preselect the patients that are more likely to receive survival benefits from anti-angiogenic therapies and to limit side effects.

An array of anti-angiogenic biomarkers including soluble vascular endothelial growth factor receptor 1 (sVEGFR1), soluble vascular endothelial growth factor receptor 2 (sVEGFR2), placental growth factor (PlGF) and VEGF are considered potential pharmacodynamic biomarkers. Their dynamics in peripheral blood samples are proposed to be associated with response to treatment and duration of survival [15].

To determine the value of these biomarkers, we were interested if reported data from GBM clinical trials could be evaluated for biomarkers of response to standard of care therapy or other treatment regimens trialled in the clinic. We therefore conducted a systematic review of key molecular biomarkers that have been investigated for their predictive value in recent GBM clinical trials and performed meta-analyses of such markers where statistical power (reported association with response) was sufficient.

## 2. Materials and Methods

### 2.1. Protocol and Registration

This review was registered in PROSPERO (registration number CRD42021238962) and was designed and carried out using Preferred Reporting Items for Systematic Reviews and Meta-analysis (PRISMA) formatting and guidelines [16].

### 2.2. Study Design and Search Strategy

A comprehensive literature search was conducted using three electronic databases: PubMed, Cochrane library and Embase databases for recent articles published between January 2010 to December 2021.The search strategy was deliberately broad and based on combination of keywords. The search terms used were “brain cancer biomarkers”, “glioblastoma biomarkers” and “glioma biomarkers”. Clinical studies published in English language in the last 10 years until December 2021 involving human subjects only were searched. Additional filters to include only clinical trials and randomised controlled trials (RCT) were applied in Pubmed and Embase. Included articles were screened for additional relevant studies cited for inclusion in our analysis if meeting criteria. The studies were then imported into the Rayyan Qatar Computing Research Institute (QCRI) systematic review application for further evaluation [17].

### 2.3. Study Selection and Criteria

In the screening process, two reviewers (H.S. and B.P.) independently screened all the imported publications in Rayyan. Studies were included if they evaluated histopathological confirmed GBM; patient number was more than 35; contained response evaluation of biomarkers; had OS/PFS/response rate and association of biomarker with OS/PFS; were an original study (RCT, cohort study or observational study). Publications were excluded if they were duplicates, reviews, letters, comments, clinical trial protocols or conference abstracts. Upon completion of inclusion and exclusion, any disagreements were resolved by consensus between the two reviewers. Included studies were inspected for duplication of patient cohorts or part of cohorts and if found to be duplicated the one with the most up to date data were included to avoid that the same data for identical cohorts was not included more than once.

### 2.4. Data Extraction

Ultimately a subset of 26 publications were included for data extraction and analysis and uploaded to Covidence for data extraction and quality assessment using the data extraction tool adapted for the current study. Extracted data included: general information (study title, lead author details), characteristics of included studies (study design, biomarkers tested, intervention and treatment outcomes (OS and PFS) associated with biomarkers, histopathology of tumour, total number of participants. Publications were included in meta-analyses if the hazard ratio and confidence intervals (HR and 95% CI) for the biomarkers affecting OS and PFS were given or were reliably calculated from provided Kaplan–Meier curves. For biomarkers, where number of studies or patient number did not warrant meta-analysis descriptive qualitative analyses was included. After detailed evaluation and discussion between two reviewers, 19 out of 26 studies were included in the meta-analyses while biomarkers of 7 studies underwent descriptive qualitative analysis.

### 2.5. Quality Assessment

Quality assessment was performed on all 26 included studies by two blinded reviewers using the Covidence Quality in Prognosis Studies (QUIPS) tool amended for the current study [18].

We assessed risk of bias across the six domains: study participation, study attrition, prognostic factor measurements, outcome measurement, study confounding, and statistical analysis and reporting [18]. Study participation was assessed for GBM histology, inclusion and exclusion criteria, adequate study participation (cohort size greater than 35), baseline characteristics (stage, grade, previous and current treatments). Study attrition included proportion of baseline samples available for biomarker analysis, reasons for not assessing samples (loss of follow up), attempts to collect information of non-assessed samples. Retrospective studies were not assessed for this domain. The prognostic factor measurements domain assessed whether the publication reported clear definition of prognostic factor. controls and methods for biomarker detection were valid and reliable. Method of measurement of prognostic factors is same for all the samples and is measured in an adequate proportion of study sample. The outcome measurement domain assessed whether the clear definition of outcome is provided and determined prior to biomarker analysis. Method of outcome measurement is reliable and valid. Outcome is assessed in adequate proportion of study sample and with the same method. The study confounding measurement domain assessed confounders measurements, including the previous and current treatments in relation to biomarkers, measured dose and duration of treatment. Statistical analysis and reporting assessed statistical tests used for biomarker expression in relation to survival outcomes. Appropriateness of the statistical tests for the data was assessed and description of the association of prognostic factors with the outcomes was reported.

### 2.6. Statistical Analysis and Data Analysis

Data retrieved from published reports underwent both quantitative and qualitative analysis. Statistical Analysis was performed using Review Manager (Review Manager–RevMan, 2020) and represented graphically. Random effect model based on the logarithm of the hazard ratio (HR) weighted by the inverse of the variance was used for combining results from the individual data. HR and CIs were used to evaluate the association of biomarkers with the OS. Statistical heterogeneity of included studies was assessed by the *I*^2^ statistics and chi-square test, and *I*^2^ value > 50% or *H*_eterogeneity_, 0.05 indicated substantial heterogeneity.

HR and CIs of multivariate analysis were selected preferentially if both univariate and multivariate analysis data was specified in the publication. In some cases, where HR and CIs were not given in the publications, they were calculated from the Kaplan-Meier curves using Enguage Digitizer software with reported methods [19].

## 3. Results

Of 1831 screened publications (1827 from database searches, 4 from in-publication citations), 26 studies were identified as eligible for inclusion in this review and analyses. The process of search, inclusion and exclusion of studies is presented in Figure 1 [16]. 26 studies met inclusion criteria reporting predictive and prognostic role of molecular biomarkers in GBM patients (Table 1). Meta-analysis was performed on clinically relevant biomarker information available for GBM patients in included publications. The main biomarkers analysed here included *MGMT* methylation (14 studies), *IDH1* mutation (5 studies) and EGFR expression/amplification (5 studies). Due to limited data for meta-analyses, association with OS of GBM patients was qualitatively evaluated for seven “circulatory biomarker” studies as well as one study with “cytokine and immune signature biomarkers”.

### 3.1. Risk of Bias Assessment and Sensitivity Analysis

The risk of bias quality assessment using QUIP tools is summarised in Table 2. Studies that have more than one domain assessed as high risk of bias were not included in the meta-analysis. Of 19 studies included in the meta-analysis, one study was assessed as high risk of bias for the study participation domain for not defining the inclusion criteria. This was still included in meta-analysis, as this was considered of low impact on analyses outcome. One study was assessed as high risk of bias for the study attrition domain. The included study was assessed as high risk of bias due to a smaller patient cohort size available for biomarker analyses (MGMT methylation was assessed for only 28 patients out of 53 included in that study). Studies included in the meta-analysis were either assessed as low risk of bias, moderate or unclear for the prognostic factor measurement domain and the confounding factors measurement domain. Six studies with high risk of bias for the outcome measurement domain were included in meta-analysis after carefully extracting the OS data and its association with biomarkers, while the other 4 studies did not have enough survival data for inclusion in the meta-analysis and are described qualitatively. All the studies included in the meta-analysis were assessed as low risk of bias for the statistical analyses’ domain.

Sensitivity analysis was performed manually in RevMan by taking out one study at a time to determine the effect of that study on the overall association of biomarkers with OS.

#### 3.1.1. Quantitative Analysis

##### MGMT Methylation

MGMT methylation data from fourteen studies, involving a total of 1231 patients with differing treatment regimens were included in the analysis for association of OS and MGMT status. The MGMT methylation status was determined in 10 out of 14 studies by methylation specific PCR [15,26,28,30,33,34,35,38,42,43,44]. Pyrosequencing was used in one study [40], and 3 studies did not report the methodology of *MGMT* methylation assessment [15,25,39].

Overall, *MGMT* methylation showed a significant association with better OS in GBM patients with a combined HR ratio of 1.66 (95% CI 1.32–2.09, *p* < 0.0001, random effect; Figure 2). Since the therapeutic intervention varied for the 14 studies, sub-group analysis based on therapy was also performed to evaluate differential association of *MGMT* promoter methylation with OS (Figure 2).

As expected, in patients treated with alkylating agents, there was a significant association of *MGMT* methylation with better OS, with a pooled HR ratio of 1.64 (95% CI 1.23–2.18; *p* = 0.0007). Another subgroup of patients was treated with TKIs (with or without alkylating agent in combination) also revealed significant association of *MGMT* methylation with OS, with a pooled HR ratio of 1.82 (95% CI 1.25–2.64; *p* = 0.002). Similar results were observed in the subgroup of patients receiving immunotherapy with or without alkylating combination, with a pooled HR ratio of 2.22 (95% CI 1.21–4.06; *p* = 0.01), (Figure 2). Sensitivity analysis was performed for two different treatment types (alkylating agents and tyrosine kinase inhibitors) by removing one study at a time. There was no change found in the overall significance of association of biomarker with overall survival (Appendix A).

##### IDH1 Mutation

Five studies investigated *IDH1* status in 541 patients (480 with *IDH1* wildtype and 61 with *IDH1* mutation) [15,29,30,34,35]. Treatments in this cohort included alkylating agents [29], TKIs [15,30] and immunotherapy in combination with alkylating agent [34]. One study did not specify the treatment [35]. *IDH1* mutation was significantly associated with longer OS in GBM patients irrespective of the therapeutic intervention. The pooled HR ratio was 2.37 (95% CI 1.81–3.12; *p* < 0.00001) (Figure 3). No significant effect on data outcome was observed after performing a sensitivity analysis (Appendix A).

##### EGFR Amplification or Overexpression of EGFR Protein

Five studies reported *EGFR* amplification and/or high expression of EGFR protein in a total of 575 patients [20,22,37,41,42]. Four studies included in the analysis investigated the association of high expression of EGFR [20,37,41,42] with OS and one study investigated the association of *EGFR* amplification with OS [22]. Treatment in this cohort included chemoradiotherapy (TMZ and radiotherapy) in 3 studies [37,41,42] and TKI with chemoradiotherapy in one study [22]. Treatment modality was not clearly defined in one study [20]. OS was not significantly associated with EGFR status, with a combined HR ratio of 1.31 (95% CI 0.96–1.79; *p* = 0.08) (Figure 4), possibly due to inadequate statistical power. Sensitivity analysis demonstrates the significant effect of one study [41] on the overall outcome on the association of EGFR with OS (Appendix A).

#### 3.1.2. Qualitative Analysis

Our broad search for molecular biomarkers in GBM produced a set of candidates that may have value in specific trial treatment settings. However, data were insufficient for a meta-analysis, and is summarised in Table 3 and briefly discussed below.

##### Tumour Immune Signature and Cytokine Signature

One study reported the molecular biomarkers associated with response to retroviral immunotherapy. Vocimagene amiretrorepvector (Toca 511) is a cancer selective, retroviral replicating vector that encodes cytosine deaminase. When administered, extended release 5-fluorocytosine (Toca FC) is converted by cytosine deaminase into the potent, short lived, chemotherapeutic agent, 5-fluorouracil, which diffuses into the tumour microenvironment from Toca 511–infected cells. Biomarkers that predicted the better clinical response to treatment in the TOCA 511/FC treated GBM patients were tumour immune signature and cytokine signature. Toca 511 and Toca FC cancer treatment has a putative mechanism of action that includes T cell–mediated antitumour immune activity, so the tumour immune signature based on the immune composition of the tumour micro-environment can potentially predict the clinical response in high grade glioma patients. Higher values of this signature indicate that more activated memory CD4^+^ T cells, more M1 macrophages, fewer resting Natural killer cells (NK cells), and fewer M0 macrophages were detected in patient tumour tissue. This signature was found to be higher in responders than in non-responders (Wilcoxon rank-sum test, *p* < 0.001) [21].

The anti-tumour immune activity of TOCA 511/FC treatment can also be measured by cytokine levels from the patient’s plasma samples. Accomando and colleagues measured a cytokine signature incorporating three cytokines (soluble E-selectin, Macrophage Inflammatory protein-1β and Interleukin-6) that were associated with the response to therapy and OS [21]. Increasing values of this cytokine signature indicate higher peak E-selectin, higher peak MIP-1β, and lower peak Interleukin-6 (IL-6) in peripheral blood during and after Toca 511 and Toca FC treatment [21]. A higher value of the signature was associated with improved survival (*p* < 0.001).

##### Circulatory Biomarkers

Eight studies included in this review reported the trial results of tyrosine kinase inhibitor therapies and the molecular biomarkers associated with response to treatment, including circulatory biomarkers. Circulatory biomarkers such as sVEGFR1, plasma PlGF and VEGF levels, and CECs are proposed as potential prognostic and predictive biomarkers in anti-VEGF therapies (Table 3). For the management of GBM which is characterised by high vascularisation and aberrantly high levels of VEGF expression, anti-VEGF therapies are being trialled [45]. sVEGFR1 is implicated as a negative regulator of the VEGF pathway and proposed as a resistance biomarker to anti-VEGF therapies in other solid cancers [46].

PlGF is another member of VEGF family, and its dynamics are now being considered as a potential pharmacodynamic biomarker to anti-VEGF therapy [47,48]. Overexpression of PlGF in preclinical models promotes tumour growth, which makes it an attractive therapeutic target [49].

Circulatory endothelial cells (CECs) are mature endothelial cells shed off the blood vessels as a result of vascular damage. Increased plasma levels of CECs are reported in cancer patients that corelate with VEGF levels. CECs may serve as a surrogate marker of anti-angiogenic activity that reflect the disease status and response to anti-angiogenic treatment [50].

Pharmacodynamics of blood based sVEGFR1, sVEGFR2, PlGF, VEGF, cytokine signature, and CECs may also be useful to monitor the target effect, tumour response and treatment outcome in response to anti-VEGF therapies and immunotherapy [15,21,22,23,24,31,32,36]. If these biomarkers indeed guide decision making to continue or terminate treatment in the early phases of a trial, benefit may be maximised.

Immunotherapies targeting the PDL1-PD1 axis have entered standard clinical practice for various solid cancers including (non-small cell lung cancer, gastric cancer, urothelial cancer, cervical cancer, and melanoma) [51,52,53,54]. Recent studies have shown the direct association of PDL1 expression with survival in GBM patients [55,56,57], although more studies are needed to evaluate benefit of immunotherapy in GBM.

## 4. Discussion

GBM is the most common and aggressive type of brain cancer and treatment options have not notably improved for decades. There are different molecular subtypes within GBM and, conceivably, targeting driver pathways or molecular “weaknesses” may lead to better patient outcomes. *MGMT* and *IDH1* are widely accepted biomarkers in the clinical context to provide prognostic or predictive information and their utilities are linked to the standard of care therapy. Here, we were interested in not only re-evaluating the utility of *MGMT* and *IDH1* but also other possible candidate biomarkers for their association with GBM patient OS in the setting of clinical trials using standard of care and other treatment modalities. Such biomarkers may add benefit to future clinical trials and better GBM patient management. Yet, perhaps not surprisingly, the best studied biomarkers, even in the clinical trials context, remain *MGMT, IDH1*, and EGFR.

*MGMT* methylation, as a prognostic and predictive biomarker of GBM, has been comprehensively studied previously [5,58]. Initially Stupp et al. provided evidence of association of *MGMT* promoter methylation to outcome in GBM patients treated with TMZ and radiation therapy versus radiation alone [3]. Further trials involving 206 GBM patients confirmed better survival outcomes in those with *MGMT* promoter methylation when treated with TMZ and radiation [59]. A previous meta-analysis which analysed 30 studies with the total of 2986 patients demonstrated *MGMT* methylation status as a prognostic factor in GBM patients showing significant association with better OS and progression free survival (PFS) for patient treatment with alkylating agents [5]. In our systematic review focusing on recent clinical trials (conducted in the last 10 years), we included 14 studies/1231 patients and investigated the association of *MGMT* promoter methylation with OS outcomes in GBM patients, irrespective of therapeutic intervention. Our analysis of *MGMT* methylation in GBM agrees with previous findings, manifesting a significant association of *MGMT* methylation with good OS in GBM patients. Interestingly, the survival benefit is not limited to patients treated with alkylating agents but was observed in all the GBM patients irrespective of treatment.

However, substantial heterogeneity was observed in the overall analysis of association of *MGMT* methylation with OS for the 14 included studies (*I*^2^ = 56%), while this was smaller (*I*^2^ = 38%) for studies focusing on alkylating agent treatments. We were also able to perform subgroup analysis based on the treatment type, and still found significant OS association with *MGMT* methylation. This observation is intriguing and suggests that while the close functional link between MGMT and alkylating agents would predict such a relationship, there may be more biological significance to MGMT methylation resulting in clinical benefits from other agents. Of the 7 studies included for the TKIs treatment group, 2 studies investigated newly diagnosed GBM patients who received TKIs therapy together with standard of care alkylating agents [26,38] and one study investigated newly diagnosed GBM patients treated with TKIs and radiation therapy [44]. The other 4 studies investigated the prognostic value of *MGMT* methylation in patients at first or second recurrence after standard therapy (chemotherapy with TMZ and radiation). In these 4 studies, patients were treated with either bevacizumab alone or in combination with other drugs [15,28,30,39]. The prognostic significance of MGMT methylation for progressive GBM patients treated with bevacizumab has been reported previously [60,61]. Wick et al. reported MGMT methylation as positive prognostic biomarker in the recurrent GBM patients treated with either bevacizumab or combination of bevacizumab and lomustine (HR: 0.48; *p* < 0.001) [60]. Similar findings were reported by Gleeson et al. with better OS observed in patients with MGMT methylated tumours as compared to those with unmethylated tumours (HR:0.61, *p* = 0.027) [61]. 

Three studies were included in the subgroup analysis of prognostic significance of *MGMT* methylation in patients receiving immunotherapy [25,34,39]. Two of these studies were conducted on newly diagnosed patients who also received standard of care along with immunotherapy [25,34] and one study enrolled patients at their first recurrence after standard treatment with TMZ and radiation [39]. This study compared the OS survival benefit in patients treated with nivolumab (PD-1 immune check point inhibitor) vs bevacizumab. No statistical difference was observed in the risk of death between groups (HR, 1.04; 95% CI, 0.83–1.30, *p* = 0.76). However, *MGMT* methylation status was prognostic in both groups. Taken together, these findings suggest MGMT methylation as strong prognostic biomarker in both newly diagnosed and recurrent GBM patients regardless of treatment intervention. However, the association of MGMT methylation with survival may still be functionally linked to alkylating agents and radiotherapy received either in parallel or prior to the trial.

IDH1/2 catalyses the reversible oxidation of isocitrate to yield α-ketoglutarate with simultaneous reduction of NADP+ to NADPH. This NADPH produced by the cells provides a cellular defence against intracellular oxidative damage [62]. *IDH1* mutations are found in approximately 12% of GBM patients [10]. Mutation in IDH1 is favourable for OS and an independent prognostic GBM biomarker [63]. Our analysis adds support to these findings [15,29,30,34,35].

EGFR, a receptor tyrosine kinase, upstream of central signalling pathways such as PI3K/AKT and RAS/RAF/MEK/MAPK pathways, is often altered in cancer [64]. Alterations and overexpression of EGFR are often linked with oncogenesis in GBM and are widely investigated in this context [65,66]. *EGFR* amplification and/or overexpression is observed in 50–60% of GBM [67,68]. Past studies which explored the prognostic significance of *EGFR* mutations, amplification and/or overexpression in GBM reported conflicting results [8,67,69,70,71]. While some studies found association of EGFR overexpression and amplification with poor prognosis [8,67], others did not find prognostic value of EGFR in GBM [69,71]. In addition, EGFR is also considered a potential target for newer therapies in GBM. However, the results from clinical trials targeting EGFR through various small kinase inhibitors (erlotinib, gefitinib, afatinib, and lapatinib) were disappointing, at least in part due to poor drug penetrance through the blood–brain barrier. However, adaptive reliance on redundant pathways to overcome EGFR inhibition has been proposed [72] in line with observations in other cancers treated with EGFR inhibitors.

In our meta-analysis, we included 5 studies with 575 patients and did not find significant association of *EGFR* amplification and or overexpression with OS in GBM patients. However, substantial heterogeneity was found among the included studies (*I*^2^ = 81%). Factors that may contribute to the heterogeneity include methods of determination of EGFR expression and amplification, therapeutic intervention, first diagnosis vs recurrence, median age, ethnic diversity and experimental design. Four studies included in this review assessed EGFR overexpression by immunohistochemistry [20,37,41,42], while amplification of the *EGFR* gene was assessed by fluorescence in-situ hybridisation (FISH) in another study [22]. Among the included studies, 4 studies investigated the overexpression of EGFR and its prognostic value in the patients receiving standard of care [20,37,41,42]. Results were diverging with some showing strong association of EGFR overexpression with worse survival while others produced no or limited association with survival [20,37,41,42].

Of note, one study investigated the association of EGFR amplification with OS in patients receiving standard of care treatment in combination with anti-VEGF TKI cediranib [22]. This study demonstrated improved survival in a subset of newly diagnosed GBM patients with improved tumour blood perfusion after receiving standard 6 weeks of fractionated radiation along with daily temozolomide and cediranib. They found an interesting correlation of EGFR amplification with lack of increase in perfusion after treatment. EGFR amplification was thus a negative prognostic factor for the patients treated with this combination therapy. Further detailed investigation is needed to determine whether EGFR is merely a poor prognostic variable or if it is associated with the vascular function after anti-VEGF therapies.

While our data based on limited patient numbers suggests no statistical association of EGFR amplification/overexpression, more homogeneous studies and larger patient cohorts are needed to clarify the prognostic and predictive significance of EGFR in GBM.

Other important biomarkers discussed in the review include the circulatory biomarkers (sVEGFR1, sVEGFR2, PlGF, VEGF, cytokine signature, and CECs). These pharmacodynamic biomarkers can be used to examine the target effect, tumour response and treatment outcome for drugs targeting tyrosine kinase receptors [47]. The closer examination of these biomarkers in the early phases of trials may be helpful in directing management decisions.

## 5. Conclusions

In conclusion, our meta-analysis confirms the positive prognostic significance of MGMT methylation and IDH1 mutation in GBM patients regardless of treatment type. The prognostic significance of EGFR amplification and overexpression still needs clarification. We also highlighted potential biomarkers, especially easily accessible circulating blood-based markers, which, however, need thorough future evaluation of their prognostic and/or predictive utility for GBM in certain therapy settings. This study also highlights the key knowledge gap in the literature which did not produce sufficient data to perform meta-analysis on the biomarkers associated with novel therapies.

## 6. Limitations

This review has several limitations that need to be considered. Firstly, we deliberately used broad search terms to retrieve all the studies evaluating prognostic and predictive biomarkers with standard of care or novel treatment modalities for GBM patients. Although considerable numbers of studies were identified in our search, a large proportion of studies were excluded due to their small cohort size (*n* < 35) and inclusion of patients with brain metastases. Secondly, insufficient data on novel biomarkers precluded a meta-analysis and we are therefore unable to provide evidence for their prognostic or predictive value.

The low number of studies included in the meta-analysis of EGFR as biomarker was another limitation of our review, so we combined EGFR amplification and overexpression to increase sample size. Further analysis with a greater number of studies and homogeneous biomarker detection is required to clarify evidence towards the prognostic significance of EGFR in GBM.

Another limitation was the inclusion of clinical trials that showed no survival benefits of trial drugs over standard of care. Thus, biomarkers evaluated in this context hold no value for prediction of response to the trial treatment over standard of care.

Finally, the variation in methodologies for molecular investigation could confound any statistical associations, either in favour of or against the trial hypothesis. Conceivably, the use of ‘better’ methods of determining molecular alterations, and optimised tissues (biopsy vs circulating) in carefully conducted trials with rigorous sampling and storage conditions, and sufficient follow-up with many longitudinal samples, even if not of large size, can provide good evidence of predictive and prognostic significance.

## Figures and Tables

**Figure 1 ijms-23-08835-f001:**
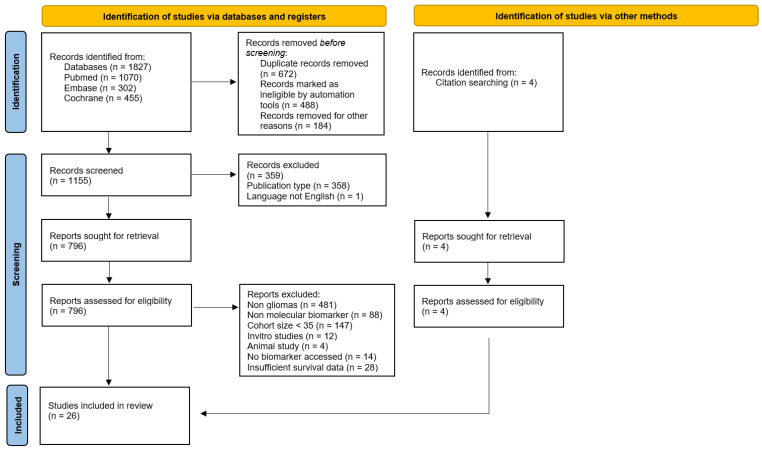
A PRISMA flow diagram of literature screening and exclusion criteria.

**Figure 2 ijms-23-08835-f002:**
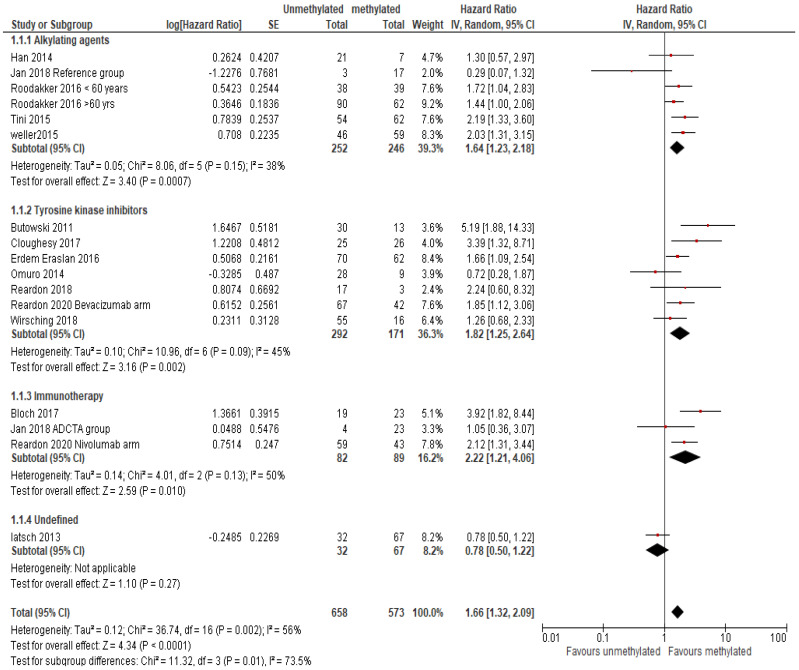
A forest plot demonstrating the association of *MGMT* methylation status with OS [15,25,26,28,30,33,34,35,38,39,40,42,43,44]. Abbreviations: SE: standard error; CI: confidence interval, bev= bevacizumab, niv= nivolumab < 60= < 60 years, > 60= >60 years. Size of the red square indicates the relative weight of the study as it contributes to the results of the overall comparison. The diamond at the bottom of the forest plot shows the result when all the individual studies are combined and averaged. The effect measure used was HR, where values greater than 1.0 indicate that patients with *MGMT* methylation has low risk of mortality than patients with unmethylated *MGMT* and vice versa for values less than 1.0.

**Figure 3 ijms-23-08835-f003:**
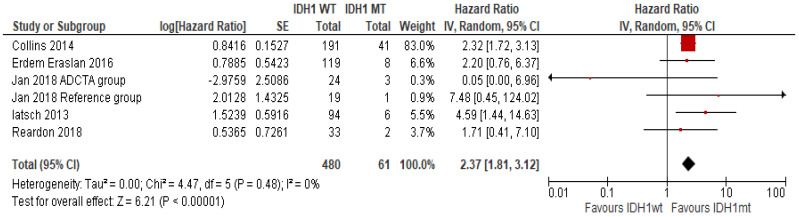
The association of OS with IDH1 mutation status [15,29,30,34,35]. Abbreviations; WT = wild type, MT = mutant. Size of the red square indicates the relative weight of the study as it contributes to the results of the overall comparison. The diamond at the bottom of the forest plot shows the result when all the individual studies are combined and averaged. The effect measure used was HR, where values greater than 1.0 indicate that patients with IDH1 MT has low risk of mortality than patients with IDH1 WT and vice versa for values less than 1.0.

**Figure 4 ijms-23-08835-f004:**
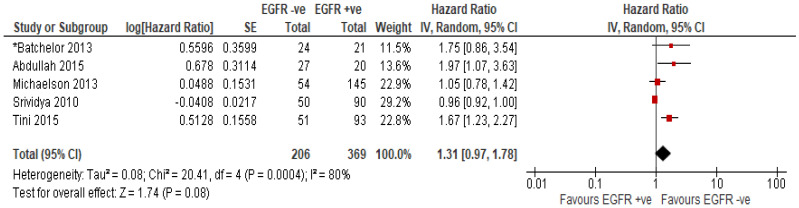
The association of OS with *EGFR* amplification or EGFR overexpression [20,22,37,41,42]. * Treatment type = chemoradiotherapy + TKI; Size of the red square indicates the relative weight of the study as it contributes to the results of the overall comparison. The diamond at the bottom of the forest plot shows the result when all the individual studies are combined and averaged. The effect measure used was HR, where values greater than 1.0 indicate that patients with *EGFR* amplification or EGFR overexpression has high risk of mortality than patients with no *EGFR* amplification or EGFR overexpression and vice versa for values less than 1.0. Note: expression of EGFR was determined by immunohistochemistry.

**Table 1 ijms-23-08835-t001:** The characteristics of included studies.

Study	Published Year	Histology	Study Design	Treatment	Median Age	No. of Patients	Endpoint/ Outcome	Biomarker Analysed
Abdullah et al. [20]	2015	Newly diagnosed GBM	R	Adjuvant chemotherapy ^α^ + Radiotherapy	83	58	OS	EGFR, TP53
Accomando et al. [21]	2020	Recurrent GBM	R	Retroviral treatment Toca 511 + Toca FC	55	56	OS	Tumour immune signature and cytokine signature
Batchelor et al. [22]	2013	Newly diagnosed GBM	RCT	TKI (cediranib) + chemoradiotherapy	57	46	OS	EGFR, PDGFRA, MET and circulatory biomarkers
Batchelor et al. [23]	2017	Recurrent GBM	Clinical trial	TKI (tandutinib)	56	56	OS	circulatory biomarkers
Beije et al. [24]	2015	Recurrent GBM	P	TKI (bev/lomustine)	57	141	OS	CECs (circulatory epithelial cells)
Bloch et al. [25]	2017	Newly diagnosed GBM	RCT	Immunotherapy (HSPPC-96Prophage) + chemoradiotherapy	58	46	OS	MGMT, PDL1
Butowski et al. [26]	2011	Newly diagnosed GBM	RCT	TKI (enzastaurin) + chemoradiotherapy	57	66	OS	MGMT
Carvalho et al. [27]	2021	Recurrent GBM	R	TKI (bev + irinotecan)	59	40	OS	c-MET, VEGFR2
Cloughesy et al. [28]	2017	Recurrent GBM	RCT	TKI Arm 1 = (onartuzumab + bev) Arm 2 = (Pla + bev)	Arm1 = 57 Arm2 = 55	Arm1 = 64 Arm 2 = 65	OS	MGMT
Collins et al. [29]	2014	Recurrent GBM	R	Alkylating agents (TMZ/PVC)	53	309	OS	IDH1
Erdem-Eraslan et al. [30]	2016	Recurrent GBM	R	TKI (lomustine/bev)	57	148	OS	MGMT, IDH1
Galanis et al. [31]	2013	Recurrent GBM	Clinical trial	TKI (bev/sorafenib)	55	54	OS	Circulatory biomarkers, CECS
Gerstner et al. [32]	2015	Recurrent GBM	Cohort study	TKI (cediranib maleate + cilengitide)	54	45	OS	Circulatory Biomarkers
Han et al. [33]	2014	Recurrent GBM	Cohort study	Alkylating agents (TMZ)	53	60	OS	MGMT
Jan et al. [34]	2018	Newly diagnosed GBM	Cohort study	Immunotherapy (ADCTA vaccine) + chemoradiotherapy	51.8 *	ADCTA = 27 Reference = 20	OS	MGMT, IDH1
Lotsch et al. [35]	2013	Newly diagnosed GBM	R	NA	60 *	100	OS	MGMT, IDH1
Lee et al. [36]	2015	Newly diagnosed GBM	RCT	TKI (vandatinib) + chemoradiotherapy	Arm1 = 55 Arm2 = 59	Arm1 = 36 Arm 2 = 70	OS	Circulatory biomarkers
Michaelsen et al. [37]	2013	Newly diagnosed GBM	P	chemoradiotherapy	59.2	225	OS	MGMT, EGFR, TP53
Omuro et al. [38]	2014	Newly diagnosed GBM	Clinical trial	TKI (bev)+ chemoradiotherapy	55	40	OS	MGMT
Reardon et al. [15]	2018	Recurrent GBM	Cohort study	TKI (trebananib/bev)	Cohort 1 = 61.9 Cohort 2 = 63.1	Cohort1 = 11 Cohort 2 = 37	OS	Circulatory biomarkers, MGMT, IDH1
Reardon et al. [39]	2020	Recurrent GBM	RCT	TKI (nivolumab/bev)	Arm 1 = 55.5 Arm 2 = 55	Arm1 = 184 Arm 2 = 185	OS	MGMT
Roodakker et al. [40]	2016	Newly diagnosed GBM	R	Chemoradiotherapy	N1 = 57 * N2 ≥ 60 N3 ≤ 60	N1 = 86 N2 = 174 N3 = 80	OS	MGMT
Srividya et al. [41]	2010	Newly diagnosed GBM	P	Chemoradiotherapy	47	140	OS	EGFR
Tini et al. [42]	2015	NA	R	Chemoradiotherapy	63	144	OS	EGFR, MGMT
Weller et al. [43]	2015	Recurrent GBM	RCT	Alkylating agents (TMZ)	Arm 1 = 58 Arm 2 = 56	Arm1 = 52 Arm 2 = 53	OS	MGMT
Wirsching et al. [44]	2018	Newly diagnosed GBM	Clinical trial	TKI (bev) + rad	70	75	OS	MGMT

Studies are labelled as the last name of the first author and presented in alphabetical order. Abbreviations: Toca 511 = Vocimagene amiretrorepvector; Toca FC = 5-fluorocytosine; TMZ = Temozolomide, rad = radiation therapy, TKI = Tyrosine kinase inhibitors, bev = bevacizumab, Pla = Placebo, PVC = (procarbazine, CCNU (1-(2-chloroethyl)-3-cyclohexyl-1-nitrosourea) and vincristine, ADCTA = autologous dendritic cell tumour antigen vaccine, chemoradiotherapy = radiation therapy + chemotherapy with TMZ; R = Retrospective study, P = prospective study, RCT = Randomised control trial, OS = Overall survival; * = mean age; # = mean + STD DEV; N1 = screening cohort, N2 and N3 = Validation Cohort. ^α^ = Chemotherapeutic drug not specified. NA = Treatment modality not given in the study.

**Table 2 ijms-23-08835-t002:** Risk of bias assessment.

Study ID	1.5 Summary of Study Participation	2.4 Summary Study Attrition	3.4 Summary of Prognostic Factor Measurement	4.4 Outcome Measurement Summary	5.3 Summary of Confounding Factors	6.4 Statistical Analysis and Reporting Summary
Abdullah 2015 [20]	Low	NA	Low	High	Moderate	Low
Accomando 2020 [21]	Low	NA	High	High	High	High
Batchelor 2013 [22]	Low	Low	Low	High	Low	Low
Batchelor 2017 [23]	Low	Unclear	Low	High	High	Low
Beije 2015 [24]	Low	Unclear	Low	Low	High	Low
Bloch 2017 [25]	Low	Low	Low	Low	Low	Low
Butowski 2011 [26]	Low	Low	Unclear	Low	Low	Low
Carvalho 2021 [27]	Low	NA	Low	Low	Low	Low
Cloughesy 2017 [28]	Low	Low	Low	High	Low	Low
Collins 2014 [29]	Low	NA	Low	Low	Low	Low
Erdem-Eraslan 2016 [30]	Low	NA	Low	Low	Low	Low
Galanis 2013 [31]	Low	Low	Low	Low	Low	Low
Gerstner 2015 [32]	Low	Low	Low	Low	Low	Low
Han 2014 [33]	Low	High	Low	Low	Low	Low
Jan-18 [34]	Low	Low	Low	Low	Low	Low
LÃ¶tsch 2013 [35]	High	NA	Low	Low	Low	Low
Lee 2015 [36]	Low	Low	Low	High	Low	Low
Michaelsen 2013 [37]	Low	Low	Low	Low	Low	Low
Omuro 2014 [38]	Low	Low	Low	High	Low	Low
Reardon 2018 [15]	Low	Low	Low	Low	Low	Low
Reardon 2020 [39]	Low	Low	High	Low	Low	Low
Roodakker 2016 [40]	Low	NA	Low	High	Low	Low
Srividya 2010 [41]	Low	Low	Low	Low	Low	Low
Tini 2015 [42]	Low	NA	Low	High	Low	Low
Weller 2015 [43]	Low	Low	Low	Low	Low	Low
Wirsching 2018 [44]	Low	Low	Unclear	High	Low	Low

Risk of bias accessed by QUIPS tool. NA = not applicable (domain not accessed for retrospective studies).

**Table 3 ijms-23-08835-t003:** The association of other biomarkers with treatment response in GBM patients.

Study	Treatment	Biomarker	Outcome
Batchelor et al. 2013 [22]	Chemoradiation + cediranib	sVEGFR1	High plasma sVEGFR1 at treatment cycle 2/day 1: poor PFS & OS (*p* < 0.05)
Batchelor et al. 2017 [23]	tanutinib	sVEGFR1, plasma PlGF	1. Decrease in sVEGFR1 at treatment cycle 2/day 1: longer PFS & OS (*p* = 0.05; 0.01 respectively) 2. Decrease in plasma PlGF at day 10: longer PFS (*p* = 0.04)
Lee et al. 2015 [36]	Chemoradiation + vandatinib	sVEGFR1, plasma PlGF	1. Longitudinal sVEGFR1 increase: poor OS (*p* < 0.05) 2. Longitudinal PlGF increase: poor OS (*p* <0.05)
Gerstner et al. 2015 [32]	cediranib maleate + cilengitide	Plasma PlGF	Early PIGF increase (at day 2): longer PFS (*p* = 0.03)
Reardon2018 [15]	trebananib/bevacizumab	Plasma VEGF and Interleukin-8 (IL-8) levels	1. High plasma VEGF: poor PFS & OS (*p* < 0.005) 2. High plasma IL-8: shorter OS (*p* < 0.05)
Beije et al. 2015 [24]	bevacizumab (avastin)/bevacizumab and lomustine/lomustine.	Circulatory endothelial cells (CECs)	For single agent lomustine treated patients with higher absolute CEC numbers after 4 and 6 weeks of treatment: longer OS (*p* = 0.03, *p* = 0.004 respectively) Absolute CEC numbers in patients receiving bevacizumab plus lomustine or bevacizumab single agent: no OS effect
Galanis et al. 2013 [31]	bevacizumab/sorafenib	Circulatory endothelial cells (CECs)	No correlation of baseline CEC values and 6 months PFS
Carvalho et al. [27]	bevacizumab	c-Met, VEGFR2	1. c-MET overexpression: TTP (*p* = 0.05) 2. VEGFR2 overexpression: Shorter TTP (*p* = 0.009) 3. Concomitant overexpression of c-Met and VEGFR2: worse TTP (*p* = 0.001) 4. Concomitant overexpression of c-Met and VEGFR2: worse OS (*p* = 0.025)
Accomando et al. [21]	Retroviral treatment Toca 511 + Toca FC	Pre-treatment tumour immune signature (in tumour microenvironment), post treatment Cytokine signature (in plasma)	1. Tumour immune signature was found to be higher in responders than non-responders (*p* < 0.001) 2. High cytokine signature: improved survival (*p* < 0.05)

Abbreviations: TTP = time to progression; Toca 511 = Vocimagene amiretrorepvector; Toca FC = 5-fluorocytosine; Tumour immune signature = Activated memory CD4 T cells * M1 macrophages/1 + Resting NK cells * M0 macrophages; Cytokine signature = E-selectin_max_ * MIP-1β_max_/1 + IL6_max_; Max = maximum value of the 3 cytokines.

## Data Availability

Not applicable.

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
