# Peer review of "Molecular Biomarkers in Glioblastoma: A Systematic Review and Meta-Analysis"

_ijms, 2022, doi:10.3390/ijms23168835_

Round 1

Reviewer 1 Report

Dear authors,

You have done a careful and extensive review. I only propose to change glioblastoma instead of glioblastoma multiforme. The term of glioblastoma multiforme is not more in use, according to the WHO classification of the central nervous system tumours 2016.

Author Response

Dear Reviewer,

Thank you very much for your comments on our manuscript.

We have made the suggested changes as described below:

“Glioblastoma multiforme” has been replaced with “glioblastoma” throughout. In the title as well abstract (page1, line 12), Keywords (page 1, line 35) and in the text (page 1, line 38).

Reviewer 2 Report

Glioblastoma multiforme (GBM) is a highly aggressive brain tumor, with a poor prognosis and an unmet medical need. Indeed the standard therapeutic approach is stationary since 2005 and clinical relapse is the rule leading the patients to a certain fatal destiny. For all these reasons finding molecular biomarkers to predict the outcome of current or new therapies is of great clinical relevance. Anyway, the meta-analysis study performed by the authors doesn’t add any novelty to the GBM knowledge.

The manuscript shows several main limitations:

·      -  The authors mainly studied three biomarkers modifications, such as MGMT methylation, IDH1 mutation and EGFR overexpression and they reviewed  their prognostic role in GBM patients by a meta-analysis. They included a small number of studies in the meta-analysis. Indeed, there are already several and larger meta-analysis studies performing a correlation of these old markers with OS of GBM patients.

·     -  The authors agree with previous studies in which correlation of MGMT methylation and/or IDH1 mutation has been extensively demonstrated. Perhaps, the only novelty is that the survival benefit is irrespective of treatment (not only alkylant agents).

·     -   Different biomarkers, in addition to those analyzed in meta-analysis, were included by the authors as a descriptive qualitative analysis. Data analysis comes from a really small number of studies and the authors don’t investigate results in the discussion section, concluding that other potential biomarkers need future evaluation for their predictive utility in GBM.

·       -  A thorough check of English usage would be helpful to make the manuscript clearer and fluider to readers.

Minor points:

·         Materials and Methods. 2.4 Data Extraction

Why do the authors mention table 3 in M&M? They also would cite tables 1 and 2.

In my opinion, they would only show tables in Result section.

·         Results. 3.1 Risk of Bias Assessment

The authors write “Six studies with high risk of bias for the outcome measurement domain were included in meta-analysis after carefully extracting the OS data and its association with biomarkers.” Indeed, there are ten studies with high risk of bias in that domain. It would be nice if they explain why they chose six studies.

·         Table 2. Risk of bias assessment

In the column of Study ID the authors missed the year in the second line (Accomando)

·         Table 2. Risk of bias assessment

In the legend the authors write they used different colors for high, low, moderate or unclear risk of bias. Instead they didn’t use colors in table 2.

Author Response

Dear Reviewer,

Thank you very much for your expert review's and comments on our manuscript. 

kindly see our responses to your comments and suggested changes as described below:

  • The authors mainly studied three biomarkers’ modifications, such as MGMT methylation, IDH1 mutation and EGFR overexpression and they reviewed their prognostic role in GBM patients by a meta-analysis. They included a small number of studies in the meta-analysis. Indeed, there are already several and larger meta-analysis studies performing a correlation of these old markers with OS of GBM patients. The authors agree with previous studies in which correlation of MGMT methylation and/or IDH1 mutation has been extensively demonstrated. Perhaps, the only novelty is that the survival benefit is irrespective of treatment (not only alkylating agents).

Response: We agree with the reviewer that it is unfortunate (but not unpredicted) that the GBM field still lacks good biomarkers. This study is important as it highlights this gap in biomarker availability. We here performed a systematic review and meta-analysis to find correlations of molecular biomarkers with GBM OS by focusing on recent publications (2010-2021) and those that have arisen from clinical trials, with data for at least 35 patients. We were reasoning that any relevant biomarkers would have made entry into clinical trial settings, that patients are followed up with strict and comparable evaluations in clinical trials (yielding more consistent data), and that including studies with >35 patients would ensure statistical power for analysis. The limited number of studies found, especially concerning “other” biomarkers (than MGMT, IDH1 and EGFR) show the need in the field to find better biomarkers as it highlights the fact that limited progress has been made in this area of research over a decade. Our manuscript is a timely reminder that GBM remains an “orphan cancer” lacking therapies to drastically improve outcomes. Biomarkers are becoming keys for diagnosis, prognosis and monitoring disease and more work is needed to find better biomarkers.

  • Different biomarkers, in addition to those analyzed in meta-analysis, were included by the authors as a descriptive qualitative analysis. Data analysis comes from a really small number of studies and the authors don’t investigate results in the discussion section, concluding that other potential biomarkers need future evaluation for their predictive utility in GBM.

Response: While we agree with the reviewer that more thorough analysis of such markers would be desirable our above outlined criteria meant smaller studies (<35 patients) were excluded during systematic evaluation. Nevertheless, some of the few included such studies indicated utility of more novel or more trial context specific markers we thought of interest. Therefore, we discussed those in the manuscript in the discussion section in a concise manner (highlighted page 15 line 471-476). We did not dwell too much on speculations though without having reliable data and hence kept discussion short. It is noteworthy, that, since small studies are less informative larger collaborative studies with larger number of patients may in the future allow to draw reliable conclusions about the association of biomarkers with OS.

  • A thorough check of English usage would be helpful to make the manuscript clearer and fluider to readers.

Response: Manuscript has been thoroughly checked again by all authors (including native English speakers A/prof Tara Roberts and Prof. Paul de Souza ).

Minor points:

  1. Materials and Methods. 2.4 Data Extraction. Why do the authors mention table 3 in M&M? They also would cite tables 1 and 2 In my opinion, they would only show tables in Result section.

Response: The recommended change has been made. The reference to table 3 in M&M is removed. (Page 3, line 134)

  1. 3.1 Risk of Bias Assessment

The authors write “Six studies with high risk of bias for the outcome measurement domain were included in meta-analysis after carefully extracting the OS data and its association with biomarkers.” Indeed, there are ten studies with high risk of bias in that domain. It would be nice if they explain why they chose six studies.

Response: The reason to include only 6 studies has been explained in the manuscript (Page 7, lines 214-215).the other 4 studies did not have enough survival data for inclusion in the meta-analysis and are described qualitatively. 

  1. Table 2. Risk of bias assessment

In the column of Study ID, the authors missed the year in the second line (Accomando)

Response: The recommended change has been made. The year to the study ID has been added. (Page 7, table 2)

  1. Risk of bias assessment

In the legend the authors write they used different colors for high, low, moderate or unclear risk of bias. Instead, they didn’t use colors in table 2.

Response: The legend in table 2 has been amended.  (Page 8 line 222)

Reviewer 3 Report

The manuscript entitled Molecular Biomarkers in Glioblastoma Multiforme: A System- 2 atic Review and Meta-Analysis addresses a topical issue in glioblastoma research area, yet it needs some improvements in order to be published.

The main weaknesses of this meta-analysis are:

-    -   they have only 2 who abstracted data given, in accordance with the checklist of a meta-analysis, should be more then 2

-     -    the number of screened articles is different in page 1, row 23 (1831) and in Figure 1 (1827)

-       - should mention the sensitivity analysis

Considering the above mentioned suggestions, we recommend the publication of the manuscript after revisions are made; still the final decision belongs to Editor-in-chief.

Author Response

Dear Reviewer,

Thank you very much for your comments on our manuscript.

Kindly find the responses to the comments and suggested changes made as described below:

  • They have only 2 who abstracted data given, in accordance with the checklist of a meta-analysis, should be more then 2

Response: We are not entirely sure we understand the issue the reviewer raises. We assume it refers to the number of investigators that extracted/evaluated data. If this is correct, we wish to highlight that this is not clearly ,mentioned in the the PRISMA guidelines (https://www.bmj.com/content/372/bmj.n71) and as common good practice for systematic reviews and meta-analyses two blinded investigators extracted data independently.

  • The number of screened articles is different in page 1, row 23 (1831) and in Figure 1 (1827)

Response: In figure 1, Records identified from databases were 1827 and records identified from citation searches were 4 which makes a total of 1831 as written in text (page  1, line 25 and page 4 line 177). We also added this for more clarity in page 4 line 177-178.

  • Should mention the sensitivity analysis

Response:  Sensitivity analysis has been performed and attached as supplementary data (table 1:MGMT methylation subgroup- alkylating agents and tyrosine kinase inhibitors; table 2: IDH1; table 3: EGFR) and is referred to in the text (page 7, line 218-220; page 8 , line 244-247; page 9, line 255-256; page 9,line 267-268; page 16, line 509).

Round 2

Reviewer 2 Report

The authors have provided a detailed answer to the comments from Reviewers and have addressed their major concerns.

The work doesn’t add significant additional informations to the biomarkers field in GBM but the revision provided by the authors is satisfactory and I have no further points or comments.